

# Towards optimized viral metagenomes for double-stranded and single-stranded DNA viruses from challenging soils

Gareth Trubl[1,4], Simon Roux[2], Natalie Solonenko[1], Yueh-Fen Li[1], Benjamin Bolduc[1], Josué Rodríguez-Ramos[1,5], Emiley A. Eloe-Fadrosh[2], Virginia I. Rich[1] and Matthew B. Sullivan[1,3]

[1] Department of Microbiology, The Ohio State University, Columbus, OH, United States of America
[2] United States Department of Energy Joint Genome Institute, Lawrence Berkeley National Laboratory, Walnut Creek, CA, United States of America
[3] Department of Civil, Environmental and Geodetic Engineering, The Ohio State University, Columbus, OH, United States of America
[4] Current affiliation: Physical and Life Sciences Directorate, Lawrence Livermore National Laboratory, Livermore, CA, United States of America
[5] Current affiliation: Department of Soil and Crop Sciences, Colorado State University, Fort Collins, CO, United States of America

Corresponding authors
Virginia I. Rich,
virginia.isabel.rich@gmail.com,
rich.270@osu.edu
Matthew B. Sullivan,
mbsulli@gmail.com

## ABSTRACT

Soils impact global carbon cycling and their resident microbes are critical to their biogeochemical processing and ecosystem outputs. Based on studies in marine systems, viruses infecting soil microbes likely modulate host activities via mortality, horizontal gene transfer, and metabolic control. However, their roles remain largely unexplored due to technical challenges with separating, isolating, and extracting DNA from viruses in soils. Some of these challenges have been overcome by using whole genome amplification methods and while these have allowed insights into the identities of soil viruses and their genomes, their inherit biases have prevented meaningful ecological interpretations. Here we experimentally optimized steps for generating quantitatively-amplified viral metagenomes to better capture both ssDNA and dsDNA viruses across three distinct soil habitats along a permafrost thaw gradient. First, we assessed differing DNA extraction methods (PowerSoil, Wizard mini columns, and cetyl trimethylammonium bromide) for quantity and quality of viral DNA. This established PowerSoil as best for yield and quality of DNA from our samples, though ~1/3 of the viral populations captured by each extraction kit were unique, suggesting appreciable differential biases among DNA extraction kits. Second, we evaluated the impact of purifying viral particles after resuspension (by cesium chloride gradients; CsCl) and of viral lysis method (heat vs bead-beating) on the resultant viromes. DNA yields after CsCl particle-purification were largely non-detectable, while unpurified samples yielded 1–2-fold more DNA after lysis by heat than by bead-beating. Virome quality was assessed by the number and size of metagenome-assembled viral contigs, which showed no increase after CsCl-purification, but did from heat lysis relative to bead-beating. We also evaluated sample preparation protocols for ssDNA virus recovery. In both CsCl-purified and non-purified samples, ssDNA viruses were successfully recovered by using the Accel-NGS 1S Plus Library Kit. While ssDNA viruses were identified in all three soil types, none were identified in the samples that used bead-beating, suggesting this lysis method may impact recovery. Further, 13 ssDNA vOTUs were identified

compared to 582 dsDNA vOTUs, and the ssDNA vOTUs only accounted for ∼4% of the assembled reads, implying dsDNA viruses were dominant in these samples. This optimized approach was combined with the previously published viral resuspension protocol into a sample-to-virome protocol for soils now available at protocols.io, where community feedback creates 'living' protocols. This collective approach will be particularly valuable given the high physicochemical variability of soils, which will may require considerable soil type-specific optimization. This optimized protocol provides a starting place for developing quantitatively-amplified viromic datasets and will help enable viral ecogenomic studies on organic-rich soils.

## INTRODUCTION

Optimization of experimental methods to generate viral-particle metagenomes (viromes) from aquatic samples has enabled robust ecological analyses of marine viral communities (reviewed in *Brum & Sullivan, 2015*; *Sullivan, Weitz & Wilhelm, 2017*; *Hayes et al., 2017*). In parallel, optimization of informatics methods to identify and characterize viral sequences has advanced viral sequence recovery from microbial-cell metagenomes, as well as virome analyses (*Edwards & Rohwer, 2005*; *Wommack et al., 2012*; *Roux et al., 2015*; *Brum & Sullivan, 2015*; *Roux et al., 2016*; *Bolduc et al., 2017*; *Ren et al., 2017*; *Amgarten et al., 2018*; *Gregory et al., 2019*). Application of these methods with large-scale sampling (*Brum et al., 2015*; *Roux et al., 2016*) has revealed viruses as important members of ocean ecosystems acting through host mortality, gene transfer, and direct manipulation of key microbial metabolisms including photosynthesis and central carbon metabolism during infection, via expression of viral-encoded 'auxiliary metabolic genes' (AMGs). More recently, the abundance of several key viral populations was identified as the best predictor of global carbon (C) flux from the surface oceans to the deep sea (*Guidi et al., 2016*). This finding suggests that viruses may play a role beyond the viral shunt and help form aggregates that may store C long-term. These discoveries in the oceans have caused a paradigm shift in how we view viruses: no longer simply disease agents, it is now clear that viruses play central roles in ocean ecosystems and help regulate global nutrient cycling.

In soils, however, viral roles are not so clear. Soils contain more C than all the vegetation and the atmosphere combined (between 1500–2400 gigatons; *Lehmann & Kleber, 2015*), and soil viruses likely also impact C cycling, as their marine counterparts do. However, our knowledge about soil viruses remains limited due to the dual challenges of separating viruses from the highly heterogeneous soil matrix, while minimizing DNA amplification inhibitors (e.g., humics; reviewed in *Williamson et al., 2017*). For these reasons, most soil viral work is limited to direct counts and morphological analyses (i.e., microscopy observations), from which we have learned (i) there are $10^7$–$10^9$ virus-like particles/g soil, (ii) viral morphotype richness is generally higher in soils than in aquatic ecosystems,

and (iii) viral abundance correlates with soil moisture, organic matter content, pH, and microbial abundance (reviewed in *Williamson et al., 2017*; *Narr et al., 2017*). The minimal collective metagenomic data for soils suggests that genetic diversity of soil viruses far exceeds that of other environments for which virome data are available and these viral communities are localized in that viruses form habitat-specific groups (*Fierer et al., 2007*; *Kim et al., 2008*; *Srinivasiah et al., 2015*; *Reavy et al., 2015*; *Zablocki, Adriaenssens & Cowan, 2016*; *Trubl et al., 2018*; *Emerson et al., 2018*; *Green et al., 2018*). Thus, while sequencing data for soil viruses is not as robust as it is in aquatic environments, such high particle counts and patterns suggest that viruses also play important ecosystems roles in soils.

The first barrier to obtaining sequence data for soil viruses is simply separating the viral particles from the soil matrix, and then accessing their nucleic acids. Viral resuspension is unlikely to be universally solvable with a single approach due to high variability of soil properties (e.g., mineral content and cation exchange capacity) impacting virus-soil interactions. There have been independent efforts to optimize virus resuspension methods tailored to specific soil types, and employing a range of resuspension methods (reviewed in *Narr et al., 2017*; *Pratama & Van Elsas, 2018*). Once viruses are separated, extraction of their DNA must surmount the additional challenges of co-extracted inhibitors (hampering subsequent molecular biology, as previously described for soil microbes; *Narayan et al., 2016*; *Zielińska et al., 2017*), and low DNA yields.

Extracting viral nucleic acid from soils typically results in very low DNA yields, requiring amplification prior to sequencing. Amplification of viral nucleic acid is necessary because the high heterogeneous nature of soil prevents any meaningful viral ecology if DNA yield is increased by increasing the number of virus extractions and pooling the concentrate (micro-scale variation and the need for smaller-scale sampling reviewed in *Fierer, 2017*). Two widely used methods to amplify viral nucleic acid are multiple displacement amplification (MDA; 'whole genome' amplification using the phi29 polymerase) and random priming-mediated sequence-independent single-primer amplification (RP-SISPA). Both allow qualitative observations of viral sequences but preclude quantitative ecological inferences. Specifically, MDA causes dramatic shifts in relative abundances of DNA templates, which impact subsequent estimates of viral populations diversity, and, most dramatically, over-amplify ssDNA viruses (*Binga, Lasken & Neufeld, 2008*; *Yilmaz, Allgaier & Hugenholtz, 2010*; *Kim, Whon & Bae, 2013*; *Marine et al., 2014*). RP-SISPA is biased towards the most abundant viruses or largest genomes, and leads to uneven coverage along the amplified genomes (*Karlsson, Belák & Granberg, 2013*). More recently, quantitative amplification methods have emerged that use transposon-mediated tagmentation (Nextera, for dsDNA; *Trubl et al., 2018*; *Segobola et al., 2018*) or acoustic shearing to fragment and a custom adaptase (Accel-NGS 1S Plus, for dsDNA and ssDNA; (*Roux et al., 2016*; *Rosario et al., 2018*)) to ligate adapters to DNA templates, before PCR amplification is used to obtain enough material for sequencing. These approaches have successfully amplified as little as 1 picogram (Nextera XT; *Rinke et al., 2016*) and 100 nanograms (Accel-NGS 1S Plus; *Laurie Kurihara et al., 2014*) of input DNA for viromes while maintaining the relative abundances of templates.
We previously optimized a viral resuspension method for three peat soil habitats (palsa, bog, and fen, spanning a permafrost thaw gradient; *Trubl et al., 2016*). Given emerging quantitative low-input DNA library construction options, we sought here to characterize how the choice of methods for viral particle purification, lysis and DNA extraction impacted viral DNA yield and quality, and resulting virome diversity. The objectives of this work were to (1) optimize the generation of viromes from soils and (2) evaluate the capability of the Accel-NGS 1S Plus kit to quantitatively amplify ssDNA and dsDNA viruses from soils. We conducted two independent experiments testing three different DNA extraction methods (Experiment 1), and then two virion lysis methods with and without further particle purification (Experiment 2). Because microscopy is not sufficient for assessing the presence of non-viral particles, we employed a combination of qPCR and virus-specific bioinformatics to evaluate the success of this protocol to yield genuine viral genomes. Quantitative soil viromes for both ssDNA and dsDNA viruses were generated, enabling a robust comparison of the different protocols tested.

## METHODS

### Field site and sampling

Stordalen Mire (68.35°N, 19.05°E) is a peat plateau in Arctic Sweden in a zone of discontinuous permafrost. Peat depth ranges from 1–3 m (*Johansson et al., 2006*; *Normand et al., 2017*). Habitats broadly span three stages of permafrost thaw: palsa (drained soil, dominated by small shrubs, and underlain by intact permafrost; pH ∼6.50), bog (partially inundated peat, dominated by Sphagnum moss, and underlain by partially thawed permafrost; pH ∼4.10), and fen (fully inundated peat, dominated by sedges, and with no detectable permafrost at <1 m; pH ∼5.70) (further described in *Hodgkins et al., 2014*). These soils vary chemically (*Hodgkins et al., 2014*; *Normand et al., 2017*; *Wilson et al., 2017*), hydraulically (*Christensen et al., 2004*; *Malmer et al., 2005*; *Olefeldt et al., 2012*; *Jonasson et al., 2012*), and biologically (*Mondav et al., 2014*; *McCalley et al., 2014*; *Mondav et al., 2017*; *Woodcroft et al., 2018*), creating three distinct habitats with increasing organic matter lability with permafrost thaw. Soil was collected with an 11 cm-diameter custom circular push corer at palsa sites, and with a 10 cm ×10 cm square Wardenaar corer (Eijkelkamp, The Netherlands) at the bog and fen sites. Three cores from each habitat were processed using clean techniques described previously (*Trubl et al., 2016*) and cut in five-centimeter increments from 1–40 cm for palsa and 1–80 cm for bog and fen cores. Samples were flash-frozen in liquid nitrogen and kept at –80 °C until processing. The sampled palsa, bog, and fen habitats were directly adjacent, such that all cores were collected within a 120 m radius. For this work, viruses were analyzed from 20–24 cm deep peat, from three cores at each of the three habitats. For Experiment 1 (DNA extraction), 18 samples were used (9 bog and 9 fen), with 10 ± 1 g of soil per sample. For Experiment 2 (virion lysis and purification), 36 samples were used (12 palsa, 12 bog, and 12 fen) with 7.5 ± 1 g of soil per sample.

## Experiment 1: optimizing DNA extraction

Viruses were resuspended using a previously optimized method for these soils (*Trubl et al., 2016*) with minor adjustments. Briefly, 10 ml of a 1% potassium citrate resuspension buffer amended with 10% phosphate buffered-saline and 150 mM magnesium sulfate was added to $10 \pm 0.5$ g peat (AKC' buffer). Viruses were physically dispersed via 1 min of vortexing, 30 s of manual shaking, and then 15 min of shaking at 400 rpm at 4 ° C. The samples were then centrifuged for 20 min at $1,500 \times g$ at 4 °C to pellet debris, and the supernatant was transferred to new tubes. The resuspension steps above were repeated two more times and the supernatants were combined, and then filtered through a 0.2 μm polyethersulfone membrane filter to remove particles and cells and transferred into a new 50 ml tube. The filtrate was then purified via overnight treatment with DNase I (Kunitz units; ThermoFisher, Waltham, Massachusetts) at a 1:10 dilution at 4 °C, inactivated by adding a final concentration of 10 mM EDTA and EGTA and mixing for 1 h. All viral particles were further purified by CsCl density gradients, established with five CsCl density layers of $\rho$ 1.2, 1.3, 1.4, 1.5, and 1.65 g/cm$^3$; we included a 1.3 g/cm$^3$ CsCl layer to collect ssDNA viruses (*Thurber et al., 2009*). After density gradient centrifugation of the viral particles, we collected and pooled the 1.3–1.52 g/cm$^3$ range from the gradient for viral DNA extraction. The viral DNA was extracted (same elution volume) using one of three methods: Wizard mini columns (Wizard; Promega, Madison, WI, products A7181 and A7211), cetyl trimethylammonium bromide (CTAB; *Porebski, Bailey & Baum, 1997*), or modified DNeasy PowerSoil DNA extraction kit (C3 reagent was 1/3 of working volume and C4 reagent was 1.5× working volume) with heat lysis (10 min incubation at 70 °C, vortexing for 5 s, and 5 min more of incubation at 70 °C) (PowerSoil; Qiagen, Hilden, Germany, product 12888). The extracted DNA was further cleaned up with AMPure beads (Beckman Coulter, Brea, CA, product A63881). DNA purity was assessed with a Nanodrop 8000 spectrophotometer (Implen GmbH, Germany) by the reading of A260/A280 and A260/A230, and quantified using a Qubit 3.0 fluorometer (Invitrogen, Waltham, Massachusetts). DNA sequencing libraries were prepared using Swift Accel-NGS 1S Plus DNA Library Kit (Swift BioSciences, Washtenaw County, Michigan), and libraries were determined to be 'successful' if there was a smooth peak on the Bioanalyzer with average fragment size of <1kb (200–800 bp ideal) and minimal-to-no secondary peak at ∼200 bp (representing concatenated adapters) (Fig. S1), and <20 PCR cycles were required for sequencing. Six libraries were successful (two from bog and four from fen) and required 15 PCR cycles. The successful libraries were sequenced using Illumina HiSeq (300 million reads, $2 \times 100$ bp paired-end) at JP Sulzberger Columbia Genome Center.

## Experiment 2: optimizing particle lysis and purification

Viromes were generated as in Experiment 1 with minor changes. First, viruses were resuspended as described for Experiment 1, except half of the samples were not purified with CsCl density gradient centrifugation. This was to follow-up on our previous work that suggested CsCl resulted in potentially a major loss of viruses (*Trubl et al., 2016*). Second, DNA was extracted from all samples using the PowerSoil method, but the physical method of particle lysis was tested by half of the samples undergoing the standard heat
lysis as above and the other half undergoing the alternative PowerSoil bead-beating step (with 0.7 mm garnet beads). Third, the extracted DNA was further cleaned up with DNeasy PowerClean Pro Cleanup Kit (Qiagen, Hilden, Germany, product 12997), instead of AMPure beads. Assessment of microbial contamination was done via qPCR (pre and post-cleanup) with primer sets 1406f (5′-GYACWCACCGCCCGT-3′) and 1525r (5′-AAGGAGGTGWTCCARCC-3′) on 5 µl of sample input to amplify bacterial and archaeal 16S rRNA genes as previously described (*Woodcroft et al., 2018*). Finally, the 12 palsa samples were sequenced at the Joint Genome Institute (JGI; Walnut creek, CA), where library preparation was performed using the Accel-NGS 1S Plus kit. All viromes required 20 PCR cycles, except –CsCl, bead-beating which required 18. All libraries were sequenced using the Illumina HiSeq-2000 1TB platform (2 × 151 bp paired-end).

## Bioinformatics and statistics

The same informatics and statistics approaches were applied to viromes from Experiments 1 and 2. The sequences were quality-controlled using Trimmomatic (*Bolger, Lohse & Usadel, 2014*), adaptors were removed, reads were trimmed as soon as the average per-base quality dropped below 20 on 4 nt sliding windows, and reads shorter than 50 bp were discarded, with an additional 10 bp removed from the beginning of read pair one and the end of read pair two to remove the low complexity tail specific to the Accel-NGS 1S Plus kit, per the manufacturer's instruction. Reads were assembled using SPAdes (*Bankevich et al., 2012*; single-cell option, and k-mers 21, 33, and 55), and the contigs were processed with VirSorter to distinguish viral from microbial contigs (virome decontamination mode; *Roux et al., 2015*).

Contigs that were selected as VirSorter categories 1 and 2 were used to identify dsDNA viral contigs (as in *Trubl et al., 2018*). ssDNA viruses, due to short genomes and highly divergent hallmark genes, can frequently be missed by automatic viral sequence identification tools (e.g., VirSorter from *Roux et al., 2015* or VirFinder in *Ren et al., 2017*). We therefore applied a two-step approach to ssDNA identification. First, we identified circular contigs that matched ssDNA marker genes from the PFAM database (Viral_Rep and Phage_F domains), using hmmsearch (*Eddy, 2009*; HMMER v3; cutoffs: score $\geq$ 50 and e-value $\leq$ 0.001). This identified four Phage_F-encoding and five Viral_Rep-encoding circular contigs, i.e., presumed complete genomes. Second, 2 new HMM profiles were generated, using the protein sequences from the nine identified circular viral contigs, and used to search (hmmsearch with the same cutoffs) the viromes' predicted proteins. This resulted in a final set of 23 predicted ssDNA contigs identified across nine viromes (Table S1).

The viral contigs were clustered at 95% average nucleotide identify (ANI) across 85% of the contig (*Roux et al., 2019a*) using nucmer (*Delcher, Salzberg & Phillippy, 2003*). The same contigs were also compared by BLAST to a pool of potential laboratory contaminants (i.e., Enterobacteria phage PhiX17, Alpha3, M13, Cellulophaga baltica phages, and Pseudoalteromonas phages), and any contigs matching a potential contaminant at more than 95% ANI across 80% of the contig were removed. Viral operational taxonomic units (vOTUs) were defined as non-redundant (i.e., post-clustering) viral contigs >10

kb for dsDNA viruses (from VirSorter categories 1 or 2; *Roux et al., 2015*) and circular contigs from 4–8 kb for Microviridae viruses or 1–5 kb for circular replication-associated protein (Rep)-encoding ssDNA (CRESS DNA) viruses. The vOTUs represent populations that are likely species-level taxa and there is extensive literature context supporting this new standard terminology, which is summarized in a recent consensus paper (*Roux et al., 2019a*; *Roux et al., 2019b*). The relative abundance of vOTUs was estimated based on post-QC reads mapping at $\geq$90% ANI and covering >10% of the contig (*Paez-Espino et al., 2016*; *Roux et al., 2019a*; *Roux et al., 2019b*) using Bowtie2 (*Langmead & Salzberg, 2012*). Figures were generated with R, using packages Vegan for diversity (*Oksanen et al., 2016*) and ggplot2 (*Wickham, 2016*) or pheatmap (*Kolde, 2015*) for heatmaps. Hierarchical clustering (function pvclust; method.dist="euclidean" and method.hclust="complete") was conducted on Bray-Curtis dissimilarity matrices using 1,000 bootstrap iterations and only the approximately unbiased (AU) bootstrap values were reported.

### Data availability

The 18 viromes from Experiments 1 and 2 are available at the IsoGenie project database under data downloads at https://isogenie.osu.edu/ and at CyVerse (https://www.cyverse.org/) file path /iplant/home/shared/iVirus/Trubl_Soil_Viromes. Data was processed using The Ohio Supercomputer Center (*Ohio Supercomputer Center, 1987*). The final optimized protocol can be accessed here: https://www.protocols.io/view/soil-viral-extraction-protocol-for-ssdna-amp-dsdna-tzzep76.

## RESULTS AND DISCUSSION

Two independent experiments were performed to optimize the generation of quantitatively-amplified viromes from soil samples (Fig. 1). Experiment 1 evaluated three different DNA extraction methods for DNA yield, purity, and successful virome generation on the challenging humic-laden bog and fen soils. Experiment 2 compared two viral particle purification methods (with or without CsCl) and two virion lysis methods (heat vs bead-beating), for DNA yield, microbial DNA contamination, and successful virome generation for all three site habitats (palsa, bog and fen). An optimized virome generation protocol was determined for these palsa, bog and fen soils.

### Experiment 1: different DNA extraction methods display variable efficiencies and recover distinct vOTUs

In Experiment 1, three DNA extraction methods were evaluated for DNA yield and purity: PowerSoil DNA extraction kits, Wizard mini columns, and a classic molecular biological approach using cetyl trimethylammonium bromide (CTAB). The PowerSoil kit was designed for humic-rich soils, which dominate our site (*Hodgkins et al., 2014*; *Normand et al., 2017*), and has performed well previously for viral samples (*Iker et al., 2013*). Wizard mini columns were used previously to generate viromes from these soils (*Trubl et al., 2018*). CTAB performs well on polysaccharide-rich samples (*Porebski, Bailey & Baum, 1997*), such as our site's peat soils.

Overall, the PowerSoil kit performed best, with the highest DNA yields and increased purity which led to more successful libraries and identification of more vOTUs in the

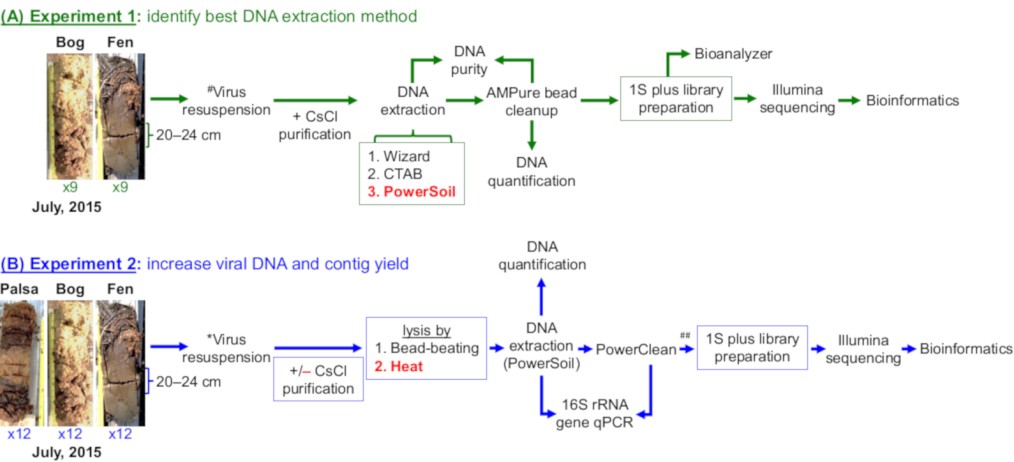

**Figure 1** **Overview of experiments to optimize methods for virome generation.** Two experiments evaluated three DNA extraction methods (A, Experiment 1 in green), two different virion lysis methods, and CsCl virion purification (B, Experiment 2 in blue), for optimizing virome generation from three peats soils along a permafrost thaw gradient. Nine soil cores were collected in July 2015, three from each habitat, and used to create 18 samples (9 bog and 9 fen) with 10 ± 1 g of soil in each sample for Experiment 1 and 36 samples (12 palsa, 12 bog, and 12 fen) with 7.5 ± 1 g of soil in each sample for Experiment 2; representative photos of cores were taken by Gary Trubl. Viruses were resuspended as previously described in *Trubl et al. (2016)*, but with the addition of a DNase step and a 1.3 g/ml layer for CsCl purification. Red font color indicates the best-performing option within each set. # denotes adapted protocol from *Trubl et al. (2016)*. ## indicates that only 12 palsa samples proceeded to library preparation.

soils tested (bog and fen). Specifically, the PowerSoil kit generally yielded the most DNA (6.34 ± 0.94 in bog and 13.64 ± 4.95 in fen), although the increase was only significant in the fen habitat (one-way ANOVA, $\alpha$ 0.05, and Tukey's test with $p$-value <0.05; Fig. 2A). DNA purity, which is also essential to virome generation (since proteins, phenols, and organics can inhibit amplification; reviewed in *Alaeddini, 2012*), was examined via A260:280 (Fig. 2B; for proteins and phenol contamination; (*Maniatis, Fritsch & Sambrook, 1982*)) and A260:230 ratios (Fig. S2; for carbohydrates and phenols; *Maniatis, Fritsch & Sambrook, 1982*; *Tanveer, Yadav & Yadav, 2016*). We posited that A260:280 is a more robust predictor of virome success, since previous work showed that A260:230 of DNA extracts had limited correlation to amplification success (*Costa et al., 2010*; *Ramos-Gómez et al., 2014*), although both are highly variable for low DNA concentrations typical for soil viral extracts. For bog samples, at least one replicate from each DNA extraction method had a clean sample based on A260:280 (defined as 1.6–2.1). For the fen, both the Wizard and PowerSoil samples were considered clean. One bog PowerSoil sample, and one fen CTAB sample, had unusually high A260:280 ratios, suggesting the presence of leftover extraction reagents in the sample.

Soil microbial metagenome protocols commonly include further DNA clean-up after extraction to remove inhibitory substances commonly seen in soil (summarized in *Roose-Amsaleg, Garnier-Sillam & Harry, 2001*; *Roslan, Mohamad & Omar, 2017*), therefore we evaluated the potential improvement in viral DNA purity from clean-up by AMPure beads. Purity (measured via A260:280) improved significantly in the bog PowerSoil + AMPure samples and was best in the CTAB + AMPure samples, while in the fen, onlyPowerSoil

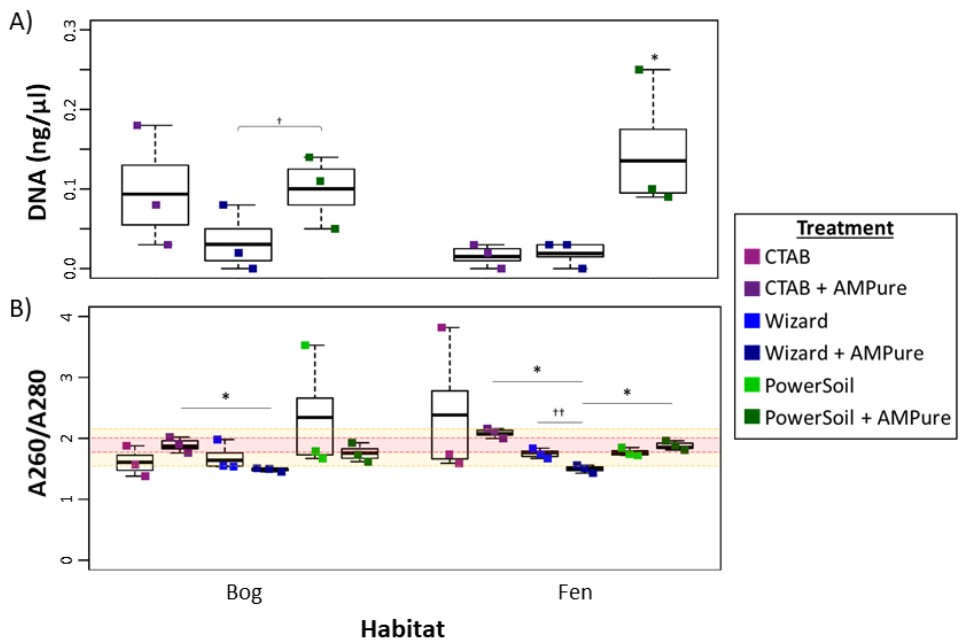

**Figure 2 Impact of extraction methods on DNA yields and purity (Experiment 1).** Bog samples are shown on the left of each panel, fen samples on the right. DNA extraction methods are color-coded: purple for CTAB, blue for Wizard, and green for PowerSoil. * denotes significant difference via one-way ANOVA, $\alpha$ 0.05, and Tukey's test with p-value < 0.05. † denotes significant difference for t test, p-value < 0.05; †† = p-value <0.01; † † † = p-value <0.001. (A) The DNA concentration (ng/μl) after AMPure purification for the three DNA extraction methods. (B) DNA extract purity via A260/A280. Dotted lines are purity thresholds: Acceptable range in yellow shading and preferred range in red shading.

extracts showed improvment. For A260:230, all post-clean-up DNAs were still below the standard minimum threshold (1.6–2.2, Fig. S2).

Although DNA extract yield and purity metrics are useful indicators of extract quality, the goal is successful library preparation and sequencing. Thus, we used the cleaned-up DNA to attempt virome generation, which revealed that PowerSoil-derived DNA was more amenable to library construction than the other extracts. Specifically, five of six PowerSoil extracts successfully generated libraries, whereas only one of the Wizard and none of the CTAB extracts led to successful library construction (threshold for success described in methods). Presumably, the success of the PowerSoil extraction methods was increased due to the kit having been optimized for humic-laden soils (specific reagents proprietary to Qiagen).

Where sequencing library construction was successful, we then sequenced and analyzed the resultant viromes to assess whether the vOTUs captured varied across replicate PowerSoil viromes and between the PowerSoil and Wizard viromes. In total, the 6 viromes produced 1,311 dsDNA viral contigs (VirSorter categories 1 and 2; *Roux et al., 2015*), which clustered into 516 vOTUs (see methods; *Roux et al., 2019a*; *Roux et al., 2019b*). There were dramatic changes in the presence and relative abundance of vOTUs across the two DNA extraction kits evaluated, the biological replicates, and the soil habitats, which is partially

the result of uneven coverage due to the 15 rounds of PCR performed to amplify the DNA (Fig. S3). While PCR amplification is a powerful tool that permits ecological interpretation of resulting viral data (*Duhaime & Sullivan, 2012*; *Solonenko & Sullivan, 2013*; *Solonenko et al., 2013*), library amplification can lead to an enrichment in short inserts, resulting in uneven coverage, a bias that scales with the number of PCR cycles performed (Roux et al. 2019). The differences in vOTU presence/absence among viromes decreased but remained noticeable even when using the most sensitive thresholds proposed for the detection of a vOTU in a metagenome (*Roux et al., 2019a*, Fig. S3). This suggests bias from the DNA extraction method (as reported previously for microbial populations; *Delmont et al., 2011*; *Zielińska et al., 2017*), and/or haphazard detection of low-abundance vOTUs due to inadequate sampling and/or sequencing depth.

## Experiment 2: heat-based lysis of non-CsCl-purified virus particles provides the most comprehensive viromes

The results of Experiment 1 identified PowerSoil as the optimal DNA extraction kit (yielding the most successful viromes), so we conducted a second experiment (Experiment 2), independent of the first, to evaluate whether density-based particle purification and/or alternative virion lysis methods could increase viral DNA yield, as previously suggested (*Delmont et al., 2011*; *Zielińska et al., 2017*). We reasoned that purification by cesium-chloride (CsCl) density gradients could result in viral loss (as previously described in *Trubl et al., 2016*), but also lead to reduced microbial DNA and particulate (e.g., clay or organic material) contamination by removing ultra-small (<0.2 um) cells, known to be present in these soils (*Emerson et al., 2018*; *Trubl et al., 2018*) or material that passes the filtration step. For lysis methods, we compared the two suggested in the PowerSoil protocol and posited that heat lysis would work better because it has been used previously on viruses (reviewed in *McCance, 1996*) and the bead-beating method was previously shown to cause ~27% more viral loss than not using beads with PowerSoil extraction kit on diverse soils (*Iker et al., 2013*).

To assess this, viruses were resuspended from three palsa, bog, and fen samples as previously described (*Trubl et al., 2016*), and then the samples were split with half undergoing particle purification via CsCl gradients and half not, and each purification treatment lysed by each of the two lysis methods (heat and bead beating) for a total of 4 treatments, all followed by PowerSoil extraction (Fig. 1). We found significant differences in DNA yield due to purification and lysis method choice (Fig. 3, one-way ANOVA, $\alpha$ 0.05, and Tukey's test with $p$-value <0.05). CsCl purification had the most impact: yield was higher without it than with it for all but one sample (Bog, –CsCl[BB]). Lysis method also mattered, with heat producing significantly higher DNA yield than bead-beating ($t$ test, $p$-value <0.05), for the –CsCl samples in the palsa and fen samples (not significant in the bog) (Fig. 3). These findings suggest that DNA yields were highest when CsCl density gradients were omitted and viral particles were lysed using heat.

Higher DNA yields could result from contaminating (i.e., non-viral) DNA, so we quantified microbial DNA in all extracts via 16S rRNA gene qPCR (Fig. 4). Surprisingly, we generally observed higher microbial contamination in the CsCl-purified samples

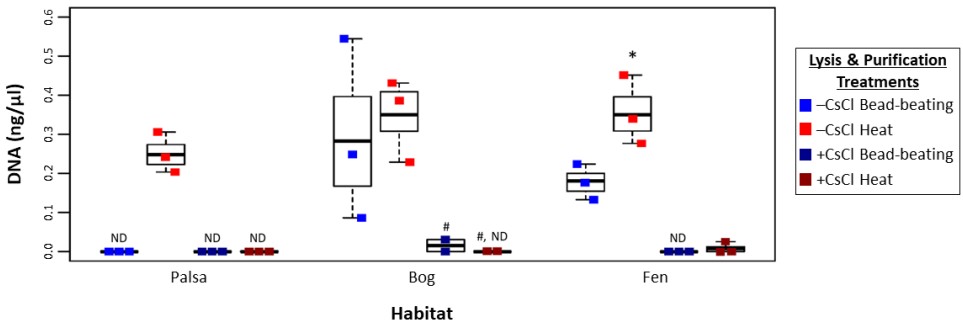

**Figure 3** **Impact of lysis and purification methods on DNA yields (Experiment 2).** The DNA concentration (ng/μl) is given for the two virion lysis methods used, with or without CsCl purification, for all three habitats. The four treatments are color coded with blue for bead-beating, red for heat lysis and a darker shade if also purified with CsCl. * denotes significant difference via one-way ANOVA, α 0.05, and Tukey's test with $p$-value < 0.05. # denotes $n = 2$. N/D denotes non-detectable DNA concentration.

(Fig. 4, one-way ANOVA, α 0.05, and Tukey's test with $p$-value <0.05), and this varied along the thaw gradient with palsa contamination being higher than that of bog and fen samples. Since residual soil organics can interfere with PCR (*Kontanis & Reed, 2006*), we repeated the qPCR assay after DNA purification with the PowerClean kit. Generally, microbial contamination increased for –CsCl samples (Fig. 4), suggesting that their previously low microbial contamination was due to PCR inhibition, and +CsCl samples had mixed results, but in each habitat +CsCl[BB] samples had a significant increase in measurable contamination (Fig. 4). All treatments had higher qPCR-based microbial contamination after PowerClean, except +CsCl[H] samples which averaged a 1.5–26-fold reduction. Overall there was still no consistent, or significant, improvement in microbial contamination from inclusion of a CsCl purification step, even after PowerClean treatment.

Since we sequenced bog and fen viromes to characterize treatment effects on the viral signal in Experiment 1, we opted in Experiment 2 to do this evaluation on the 12 palsa samples, which were all sequenced. We found that the higher DNA yields in the –CsCl samples led to ∼3-fold more viral contigs, which were also an average of 2.3-fold larger than +CsCl samples (Fig. 5A). The results from heat-lysis samples were more modest as they resulted in only ∼33% more viral contigs, and statistically indistinguishable contig sizes across treatments (Fig. 5B; unequal variance $t$-test, $p$-value >0.05). These findings suggest that the optimal combination for recovering virus genomes from these soils may be to skip CsCl purification, but still using some form of purification method (DNase used here), and lyse the resultant viral particles using heat.

We next evaluated whether vOTU representation and diversity estimates from the same samples varied across the purification and lysis methods tested here. DNA quantification of 9 out of the 12 viromes showed non-detectable amounts of DNA, but we identified vOTUs in each of the 12 palsa viromes, suggesting the Accel-NGS 1S Plus kit amplifies DNA from the picogram range. In total, 66 vOTUs were identified with 100% of the vOTUs identified in –CsCl samples, 89% (59) identified in the +CsCl samples, and vOTUs identified by both datasets displaying an average of 30-fold more coverage (Fig. 6) in –CsCl viromes. This

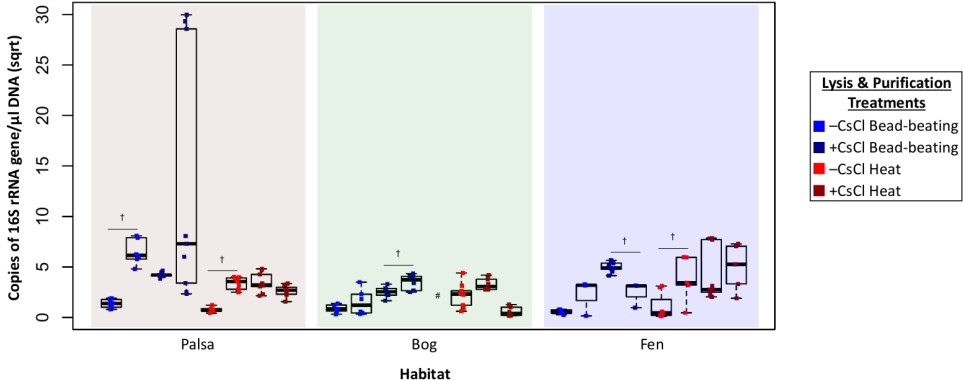

**Figure 4** **Evaluation of microbial contamination (Experiment 2).** The 16S rRNA gene contamination (square root) is indicated for each virome grouped by habitat before (left) and after (right) clean up with PowerClean. The four treatments are color coded with blue for bead-beating and red for heat lysis and a darker shade after CsCl purification. # denotes no data available. 16S qPCR primers were 1406F-1525R, from *Woodcroft et al. (2018)* [†] denotes significant difference for *t* test, *p*-value < 0.05; ††, *p*-value < 0.01; †††, *p*-value < 0.001.

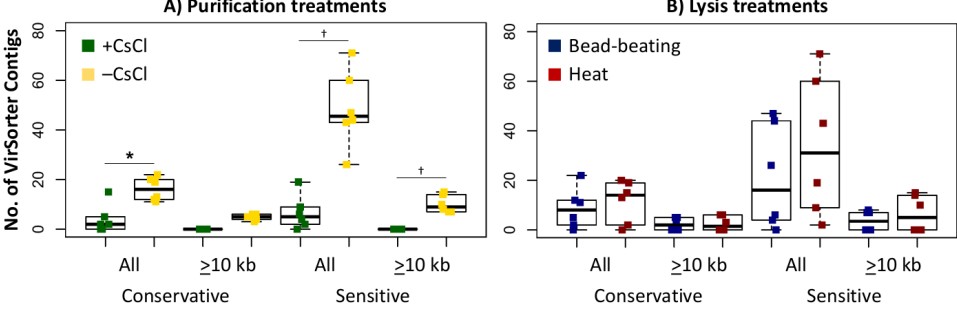

**Figure 5** **Number and size of assembled viral contigs (Experiment 2).** Boxplots show the number of viral contigs assembled, and those > 10 kb, for each treatment. Viral contigs were identified by two approaches: the "conservative" one included only contigs in VirSorter categories 1 & 2 for which a viral origin is very likely, while the "sensitive" one also included contigs in VirSorter category 3, for which a viral origin is possible but unsure.

indicates that the CsCl purification step reduced the samples to a subset of the initial viral community, it did not help recover virus genomes that would be missed otherwise, and confirmed that the 16S rRNA gene copies identified from the qPCR analyses were likely microbial contamination and not the result of 16S rRNA gene copies carried by viruses (*Ghosh et al., 2008*). Profiles of the recovered communities clustered first by soil core (AU branch supports >76), then mostly by purification (AU branch supports >66), and lastly by lysis, and did not change after varying the threshold for considering a lineage present (Fig. S4). Collectively this suggests that differences introduced by sample preparation were outweighed by the distinctiveness of each core's viral community. We proceeded to use
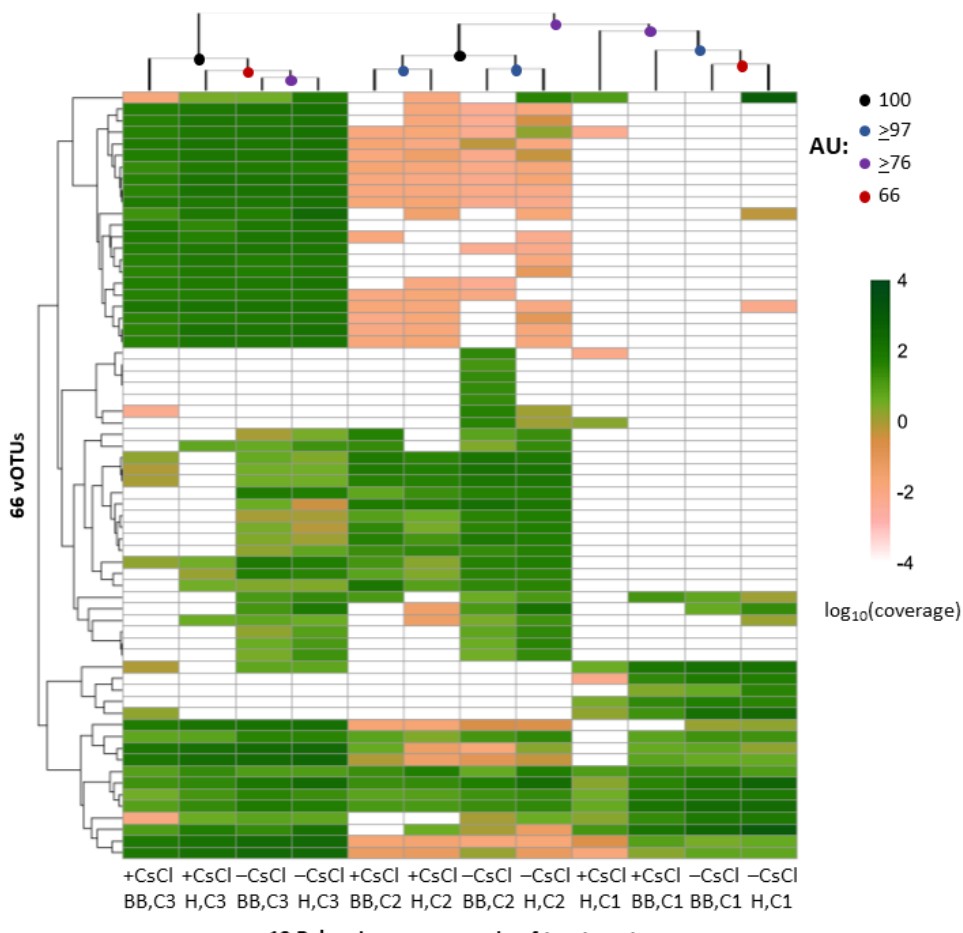

**Figure 6 Relative abundance of vOTUs across 12 palsa viromes (Experiment 2).** A heatmap showing the Euclidean-based hierarchical clustering of a Bray-Curtis dissimilarity matrix calculated from vOTU relative abundances within each virome with an approximately unbiased (AU) bootstrap value ($n = 1,000$). The relative abundances were normalized by contig length and per Gbp of metagenome and were $\log_{10}$ transformed. Reads were mapped to contigs at $\geq$90% nucleotide identity and the relative abundance was set to 0 if reads covered < 10% of the contig. Heatmaps with alternative genome coverage thresholds are presented in Fig. S3. Abbreviations: H, heat lysis; BB, bead-beating; +/– CsCl, with or without cesium chloride purification; C, soil core.

diversity metrics to evaluate the different methods' impacts. The alpha diversity metrics paralleled treatment DNA yields where –CsCl samples were on average 56% more diverse than the +CsCl samples, and heat samples were on average 83% more diverse than the bead-beating samples (Fig. S5A). A comparison of dissimilarities among samples suggested the lysis method had more of an impact, although this effect was variable between samples and thus not statistically significant overall (Fig. S5B).

**ssDNA viruses are recovered in all 3 habitats**

Previous viromic studies have been limited to describing dsDNA viruses or using MDA to describe ssDNA viruses, but with the onset of the Accel-NGS 1S Plus kit, we leveraged

the quantitatively-amplified viromics data produced here to investigate the diversity and relative abundance of ssDNA viruses in our soil samples. Culture collections have revealed ssDNA viruses commonly infect plants as opposed to bacteria, but their distributions in soils remain poorly explored outside a handful of papers which suggest they are highly diverse (*Kim et al., 2008*; *Reavy et al., 2015*; *Green et al., 2018*). Notably, the first quantitative ssDNA/dsDNA viromes suggested that identifiable ssDNA viruses represent a few percent of the viruses observed in marine and freshwater systems (*Roux et al., 2016*).

To assess this biological signal in soils, we investigated the recovery and relative abundance of ssDNA viruses across our different soil habitats and sample preparations. Overall, we identified 35 putative ssDNA viruses, 11 from the Microviridae family and 24 CRESS DNA viruses (Fig. 7), which clustered into 13 vOTUs (3 Microviridae and 10 CRESS DNA). These ssDNA vOTUs were only a small fraction of the total vOTUs identified in each habitat (1% in bog and fen, and 8% in palsa) and only bog and fen samples included both types (Microviridae and CRESS-DNA), while palsa samples included exclusively CRESS-DNA viruses (Table S1). This suggests that, as for dsDNA viruses, the composition of the ssDNA virus community varies along the thaw gradient, potentially as a result of known changes in the host communities (*Trubl et al., 2018*), both microbial (*Mondav et al., 2017*; *Woodcroft et al., 2018*) and plant (*Hodgkins et al., 2014*; *Normand et al., 2017*). Notably, bead-beating-lysis samples did not include any ssDNA viruses. We posit that this was likely due to the heterogeneity of soil, because ssDNA viruses have previously been identified from experiments that used a bead-beating lysis (*Hopkins et al., 2014*). Finally, ssDNA viruses represented on average 4% of the community in the samples where ssDNA and dsDNA viruses were detected, which suggests that ssDNA viruses are not the dominant type of virus in these soils.

## CONCLUSIONS

The development of a sample-to-sequence pipeline for ssDNA and dsDNA viruses in soils is crucial for characterizing viruses and their impact in these ecosystems. Our work here built upon previous work that optimized virus resuspension from peatland soils by evaluating DNA extraction and lysis methods to increase DNA yields and purity. Additionally, this is the first evaluation of the Accel-NGS 1S Plus kit to capture ssDNA viruses in soils and our data suggests it is also capable of amplifying DNA down to the picogram range. Although these efforts have made inroads towards characterizing the soil virosphere, several challenges remain. Initial challenges arise from resuspension and enumeration of "fake" virus particles (*Ackermann & Tiekotter, 2012*; *Forterre et al., 2013*), the lack of data on what fraction of the free virus particles are being recovered from soils, and how to achieve a holistic sampling of the virus community (i.e., dsDNA, ssDNA, and RNA viruses). After viruses are resuspended from soils and their nucleic acid is extracted, there is still a need for amplification which can cause downstream issues (e.g., uneven coverage). Beyond these, the presence of non-viral DNA in capsids or vesicles, e.g., gene transfer agents, can dilute the viral signal in metagenomes and complicate interpretation (reviewed in *Roux et al., 2013*; *Hurwitz, Hallam & Sullivan, 2013*; *Lang & Beatty, 2010*), although new methods

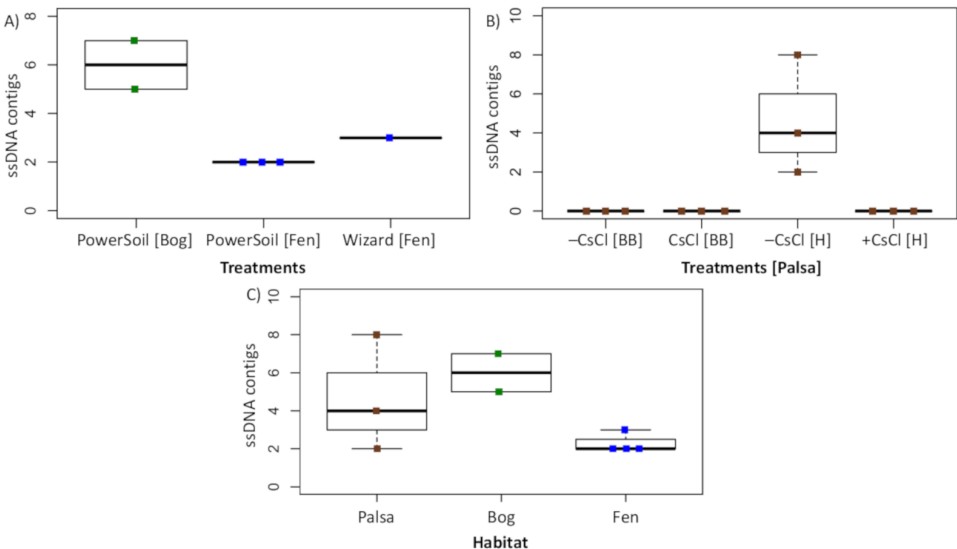

**Figure 7** **Recovery of ssDNA viruses across habitats and methods.** (A) ssDNA viral contigs from viromes in Experiment 2. The PowerSoil bog samples are grouped, as are the PowerSoil fen samples. The single Wizard virome from the fen habitat is also shown. (B) ssDNA viral contigs from viromes in Experiment 2 grouped by the four treatments: +/− CsCl and bead-beating [BB] or heat [H] virion lysis method. (C) ssDNA viruses from both Experiments are shown and grouped by habitat.

are being developed to identify and characterize these contaminating agents (reviewed in *Lang, Westbye & Beatty, 2017*). Given all the known contaminants that can pass through filtration and their unknown densities or impact on DNA extraction and amplification, we caution the removal of the CsCl purification without further assessment on additional soils.

In addition to optimization of methods to characterize soil viruses, there are many techniques that can be implemented that will greatly advance our knowledge of viruses in soils. The advent of long-read sequencing technologies have recently been applied to viromics and can improve contig generation for regions of genome with high similarity or complexity (summarized in *Roux et al., 2017*; *Karamitros et al., 2018*) and prevent formation of chimeric contigs. Longer-read viromes can thereby not only increase vOTU recovery but also provide resolution of hypervariable genome regions with niche-defining genes, and help capture micro-diverse populations missed by short-read assemblies (*Warwick-Dugdale et al., 2019*). Next, inferences of viral impacts on microbial communities and C cycling will require predicting hosts both *in silico* (*Edwards et al., 2015*; *Paez-Espino et al., 2017*) and in vitro (*Deng et al., 2014*; *Brum & Sullivan, 2015*; *Cenens et al., 2015*), approaches to which are emerging. Finally, identification of the active viral community and characterization of their roles in biogeochemical processes can be better resolved with techniques like stable isotope-based approaches linked with nanoscale secondary ion mass spectrometry (NanoSIP; *Pacton et al., 2014*; *Pasulka et al., 2018*; *Gates et al., 2018*). Application of these and other approaches to soil viromics will increase and diversify
publicly available viral datasets, advance our understanding of soil viral ecology, and improve our knowledge of viral roles in soil ecosystems.

## ACKNOWLEDGEMENTS

We thank Michelle Carlson from Qiagen for her support and technical advice. We thank Olivier Zablocki for his suggestions and comments on the manuscript. We thank Moira Hough, Sky Dominguez, and Nicole Raab for collecting the soil cores and geochemical data, and the Abisko Naturvetenskapliga Station for field support.

### Funding

This study was funded by the Genomic Science Program of the United States Department of Energy Office of Biological and Environmental Research (grants DE-SC0010580 and DE-SC0016440), The Office of Science, Office of Workforce Development for Teachers and Scientists, Office of Science Graduate Student Research (SCGSR) program, and by the Gordon and Betty Moore Foundation Investigator Award (GBMF#3790 to Matt Sullivan). The SCGSR program is administered by the Oak Ridge Institute for Science and Education (ORISE) for the DOE. ORISE is managed by ORAU under contract number DE-SC0014664. The work conducted by the US Department of Energy Joint Genome Institute, a DOE Office of Science User Facility, is supported by the Office of Science of the US Department of Energy under contract no. DE-AC02-05CH11231. Bioinformatics were supported by The Ohio Supercomputer Center and by the National Science Foundation under Award Numbers DBI-0735191 and DBI-1265383; URL: www.cyverse.org. The funders had no role in study design, data collection and analysis, decision to publish, or preparation of the manuscript.

### Grant Disclosures

The following grant information was disclosed by the authors:
Genomic Science Program of the United States Department of Energy Office of Biological and Environmental Research: DE-SC0010580, DE-SC0016440.
The Office of Science, Office of Workforce Development for Teachers and Scientists, Office of Science Graduate Student Research (SCGSR).
Gordon and Betty Moore Foundation Investigator Award: GBMF#3790.
Oak Ridge Institute for Science and Education (ORISE): DE-SC0014664.
Office of Science of the US Department of Energy: DE-AC02-05CH11231.
Ohio Supercomputer Center and by the National Science Foundation under Award Numbers: DBI-073519, DBI-1265383.

### Competing Interests

The authors declare there are no competing interests.

## Author Contributions

- Gareth Trubl, Yueh-Fen Li conceived and designed the experiments, performed the experiments, analyzed the data, prepared figures and/or tables, authored or reviewed drafts of the paper, approved the final draft.
- Simon Roux, Benjamin Bolduc and Josué Rodríguez-Ramos analyzed the data, authored or reviewed drafts of the paper, approved the final draft.
- Natalie Solonenko conceived and designed the experiments, performed the experiments, authored or reviewed drafts of the paper, approved the final draft.
- Emiley A. Eloe-Fadrosh contributed reagents/materials/analysis tools, authored or reviewed drafts of the paper, approved the final draft.
- Virginia I. Rich and Matthew B. Sullivan conceived and designed the experiments, contributed reagents/materials/analysis tools, authored or reviewed drafts of the paper, approved the final draft.

## Data Availability

Raw data is available in the IsoGenieDB, in the Viral Sequencing Data section at https://isogenie-db.asc.ohio-state.edu/datasources#viral_sequencing. All datasets named Trubl Soil Viromes were used in this study.

## Supplemental Information

Supplemental information for this article can be found online at http://dx.doi.org/10.7717/peerj.7265#supplemental-information.

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
