# Peer review of "Towards optimized viral metagenomes for double-stranded and single-stranded DNA viruses from challenging soils"

_PeerJ, doi:10.7717/peerj.7265_

## Round 0.1 · original submission · Major Revisions

I think this is a really important manuscript as there are very few investigations on viruses from soil habitats. The tested methods here could also be taken into account in sediment studies, which are equally challenging habitats to work with, if you are interested in viral metagenomes. One reviewer is very positive about the overall work, but has a number of minor comments, mostly of clarification nature. However, reviewer #1 is far more critical and also asks for a number of issues to be better clarified. Mostly important, there are a number of comments about the validity of the results that require further explanation and potentially further comparison with the literature. I understand that further experiments may not be possible in order to satisfy all the issues raised by reviewer #1. However, as a minimum, you may want to clarify those issues or stress alternative explanations for the results found in the paper.

Reviewer 1 ·

Basic reporting

Language was in English and mostly clear throughout. There were a few places where word choice or phrasing could be modified to increase clarity or reduce ambiguity:

l. 29; “sequencing data for soil viruses are hard to come by…” I do not think this is an accurate characterisation, as several (at least 5) soil metagenomes exist in public databases. Thus sequence information for soil viruses is available and accessible. I think the case could be made that comparatively less has been done with respect to soil viruses as opposed to, say, marine viruses – but this is a different message.

l. 41: “While little empirical data are available for inhibitors in soil viral extractions” I am not sure what you mean here. That few suggestions are available in the literature for dealing with or removing inhibitors? Please clarify.

l. 266-267: “These findings suggest that DNA yields are best when not purifying the resuspended viral particles and when lysed using heat.” This phrasing was difficult for me to follow. How about, “…DNA yields were highest when CsCl density gradients were omitted and viral particles were lysed using heat”?

Some important literature references have been left out:
l. 24: I found this potentially misleading, as more is known about viruses in soils than the few facts presented here. Viruses are dynamic and responsive to change (Srinivasiah et al. 2015. Dynamics of autochthonous soil viral communities parallels dynamics of host communities under nutrient stimulation. FEMS Microbiol. Ecol. 91: fiv063)

Several soil viral metagenomes have been analysed:
Fierer, N., Breitbart, M., Nulton, J., Salamon, P., Lozupone, C., Jones, R., Robeson, M., Edwards, R.A., Felts, B., Rayhawk, S., et al. (2007). Metagenomic and small-subunit rRNA analyses reveal the genetic diversity of bacteria, archaea, fungi, and viruses in soil. Appl. Environ. Microbiol. 73, 7059–7066.
Green, J.C., Rahman, F., Saxton, M.A., and Williamson, Kurt E. (2017). Quantifying aquatic viral community change associated with stormwater runoff in a wet retention pond using metagenomic time series data. Accepted MS ID: 201703009.
Kim, K.-H., Chang, H.-W., Nam, Y.-D., Roh, S.W., Kim, M.-S., Sung, Y., Jeon, C.O., Oh, H.-M., and Bae, J.-W. (2008). Amplification of uncultured single-stranded DNA viruses from rice paddy soil. Appl. Environ. Microbiol. 74, 5975–5985.
Reavy, B., Swanson, M.M., Cock, P.J.A., Dawson, L., Freitag, T.E., Singh, B.K., Torrance, L., Mushegian, A.R., and Taliansky, M. (2015). Distinct Circular Single-Stranded DNA Viruses Exist in Different Soil Types. Appl. Environ. Microbiol. 81, 3934–3945.
Zablocki, O., Adriaenssens, E.M., and Cowan, D. (2016). Diversity and Ecology of Viruses in Hyperarid Desert Soils. Appl. Environ. Microbiol. 82, 770–777.

The collective metagenomic data for soils suggest that genetic diversity of soil viruses far exceeds that of other environments for which virome data are available. It also suggests that viromes are localized – that viruses break out into habitat-specific groups.
I think it is important to mention these references and observations in order to provide a more complete view of our current understanding of viruses in soils.

I think the language regarding ssDNA virus metagenomics needs to be revised:
l. 318- 319: This is a bit disingenuous, as it seems to suggest that previous researchers were somehow negligent – when in fact, methods that specifically target library construction of ssDNA viruses have only recently been developed! Unless you want to count GenomiPhi, which has been around considerably longer, but whose selective amplification of ssDNA templates is more of an unwanted byproduct than a deliberate feature.

l. 322: Several soil viromes have been published featuring ssDNA viruses: Kim et al (2008), Reavy et al (2015), Green et al (2017), see suggested list of additional citations, above.

Experimental design

l. 34: pH and soil organic matter are likely to be important factors as well. While not an explicit part of your experimental design, I think it could/should be acknowledged that they are likely to play a role.

l. 42: one seemingly obvious solution has been omitted: why not just perform more virus extractions and then pool and concentrate the extracts until sufficient DNA has been obtained for sequencing? Such an approach would avoid all issues described from MDA and SISPA. Later on, in the methods (lines 88 – 90), it is described that 7-g to 10-g soil samples were used for DNA extractions. Is there any limitation on extracting larger masses of soil, or multiple 10-g replicates and then pooling the extracts? Why couldn't this be part of the experimental design?

l. 67: what is/are the specific hypotheses being tested? Part of the explicit reviewer instructions for the journal ask me to check that the “Research question well defined, relevant & meaningful. It is stated how research fills an identified knowledge gap.” I think the specific objective(s) of this work could be made more clear. Right now it seems to be to determine what happens if virus particle extraction conditions are changed – is that correct? If not, how would you make it more clear to your reader?

l. 91 and l. 124: I am not sure what the difference is between “optimizing DNA extraction” and optimizing particle lysis and purification,” the two different headers for these subsections. It appears to me that both subsections are trying to do the same thing – elute virus particles from soil samples and extract DNA from these particles for metagenomic library construction. But the different labels would appear to suggest that different things are happening? Could you please clarify?

l. 125 – 133: could you provide any rationale for these changes? For example, why omit CsCl density gradient centrifugation as a means of separating virus particles from potential co-eluted contaminants? Were these random adjustments to “see what happens” or was there a rationale behind these choices? Please make your reasoning for these adjustments more transparent to the reader. (I see that this is addressed in the Results/Discussion section. You could consider providing an explanation in the Methods either instead, or as well, to reduce reader confusion.)

l. 139: these libraries were sequenced at 151 bp, paired end, while the libraries sequenced from the previous section were done at 100 bp, paired end. I was highly concerned that this additional variable would affect your outcomes, making it more difficult to say for certain which factors account for differences in outcomes. I had done quite a bit of reading into the manuscript before determining that the steps done in the first part of your methods were not being directly compared to the steps done in the second part. This goes back to the headers (lines 91 and 124) and lack of clarification as to what is being done in these two sections and how they differ. I am still not entirely clear on what is different in the ultimate objectives of these two subsections so any way you can distinguish the two sections and how their goals might differ would be extremely helpful to my understanding.

l. 244: could this uneven coverage arising from PCR be avoided if we simply put more effort into obtaining a larger amount of starting material (and avoiding amplification altogether)? (see previous question about this, as well)

l. 260: loss of virus particles in CsCl density gradient centrifugation. Were different density layers tested or optimized? As mentioned in Thurber’s protocol paper (Thurber, R.V., Haynes, M., Breitbart, M., Wegley, L., and Rohwer, F. 2009. Laboratory procedures to generate viral metagenomes. Nat. Protoc. 4, 470–483, cited in this manuscript) some care must be taken in selecting density gradients, depending on the particle distribution (and genome types) of a particular sample. Thus, this may not be an insurmountable problem. In addition to cells that may pass through 0.22 um filtration, CsCl density gradients provide additional benefits by removing particulate contaminants (clay particulates, for example, have no lower limit on their dimensions, but differ greatly from virions in their density). So omission of the CsCl step seems to have an incompletely characterised cost-benefit sheet. I think it would bolster your results to try to determine more about what is gained vs. what is lost when CsCl is used (or not) in the purification of virus particles from soils.

Validity of the findings

While I do not doubt the authors observed what they reported, I think the paper would benefit overall from some healthy self-skepticism and careful consideration of the limitations of the experimental design:

l. 195: I think this is an important caveat of this paper. This work is impressive, to construct metagenomic libraries to determine how different methods skew the perception of the metagenome composition. But we can really only say that these impacts occur in the specific soils sampled and analysed. It is not clear to what extent, if any, these results may be generalised for other soils. And not to denigrate the work already done here, but each method is represented by only one library/virome and this does not enable us to differentiate within-group from across-group variation. In other words, in spite of the great amount of work done here, each specific treatment has N=1. While I would not go so far to say that it negates and findings, it is a limitation of the design that should at least be acknowledged.

l. 207: only bog and fen soils were tested. So it might be more accurate to say, “in the soils tested (bog and fen).” I am curious, why not the palsa soil? Was there a rationale behind this choice or was it more due to external constraints?

l. 210-221: it is probably worth noting that A260 reading for viral nucleic acids could be tricky, as the absorbance-to-DNA concentration factor varies depending on dsDNA, ssDNA, and RNA, and virus capsids are likely to have a mix of these different nucleic acid types. Further, there are many compounds that can interfere with spectrophotometric quantitation of nucleic acids, and many sequencing facilities favour fluorometric quantitation (e.g., PicoGreen) over spectrophotometry for this reason. I am curious why fluorometry was not used in this case. I think it would be reasonable to at least account for the strengths and weaknesses of these approaches, and to consider the possible caveats of the results as computed using spectrophotometry here.

l. 231: good point. In the end, if you can successfully generate sequence, does it matter if the A260/A280 ratio is not optimal? Still, part of the end-goal of this paper seems to be to determine how these factors (e.g., differences in DNA clean-up) affect perceptions about viral metagenome richness and composition. So I guess we will find out if it matters.

l.252: could you please provide more explanation here about how you are assigning the variance to the DNA extraction method and how you can be sure that it is not explained by amplification bias? It would seem that these two factors are entangled and I am not sure how one assigns a greater role to a particular factor over another. Any help here would be appreciated.

l. 279: I am not completely convinced by 16S PCR results alone as the definitive check for cellular contamination in your DNA preps. This is a helpful metric, and should be done, but transducing phage have been shown to carry copies of the 16S gene from bacterial hosts. Careful screening of your viral extracts under fluorescence microscopy or TEM could (perhaps should?) be used to bolster your 16S PCR results, and can quickly reveal potential cellular contamination in virus extracts. While inclusion of a CsCl step will undoubtedly result in some losses of your target viruses, it certainly reduces sources of non-viral DNA – both cellular and extracellular. I am dubious of omitting the CsCl step, which has been part of standard protocols for viral metagenome preparation for over a decade, with good reason. The simplest explanation for the increased yield for these (-)CsCl samples is the inclusion of non-viral DNA.

l. 278-290: as shown here, a perceived low contamination based on low 16S PCR copy number could easily be due to impure samples that hamper efficient PCR – and not, in fact, due to low contamination levels at all. Paradoxically, the contamination, itself, causes the samples to appear less contaminated! At least using a PCR-based method. Again, bolstering this with a secondary detection method such as microscopy would help reduce doubts here.

l. 293-294: a natural question here is: if the additional DNA gained by omitting the CsCl step is NOT viral, then how were additional contigs obtained? Or, in other words: Since you got more contigs by omitting CsCl step, you must have gotten more viral DNA this way. This is a fair interpretation, but other possibilities should be considered, and ruled out if possible. Much of this interpretation comes down to the specifics of your algorithm. It has been observed that, due to coevolution, many virus sequences mimic those of their hosts, down to G+C content and even homologous genes. So if you have contaminating host DNA in a (-)CsCl DNA extract, its removal from bioinformatic analysis is only as good as the specificity of sequence-sorting algorithm. One way to test this is to spike your viral DNA extracts with varying concentrations of prokaryotic DNA and determine when you can no longer parse host from virus: how much contaminating DNA is needed to fool your algorithm? Have such controls been performed, and perhaps reported elsewhere? (Indeed, I am wondering: if the VirSorter program is able to parse virus sequence from other microbial sources, why go through all the trouble to generate virus-enriched samples for analysis? Couldn’t you just extract total DNA and pull the virus sequences from the resulting metagenomes, using deep sequencing to get at rare taxa?) I don't wish come across as if I am pulling this all apart! I do wish to recognize the amount of effort that’s gone in here and the need to understand how our perceptions of microbial diversity may be biased by our methods. But I also believe that, as scientists, we must take a critical eye to our own work and take pains to consider other explanations, not simply the most convenient ones. And I believe the manuscript would benefit from a more critical analysis in this regard.

Additional comments

l. 32- 34: This seems an accurate characterization. But it begs the question: will the results you obtain here be applicable to any other soils, save the specific ones being analysed here?
l. 304- 306: this is an interesting finding. Different methods generate subsets rather than different sets with different degrees of overlap.

The key finding here is that omission of a CsCl step may enhance recovery and assembly of viral sequences in soil metagenomes. Since CsCl purification has been a central piece of viral metagenome library construction for over a decade, this strikes me as a bold claim requiring considerable support. Part of the challenge here is that no one knows what the viromes “should” look like, what their true structures are. While it is convenient to adopt the position that the method that gives us “the most” must be “the best,” there are reasons why this might not always be the case. For example, inclusion of non-viral sequences interpreted as an increase in viral richness. Is it possible to conceive of an experimental design wherein the true viral community composition is known, and therefore, you can determine how close your methods got you to the actual answer – a kind of control?

Related to this, it is not at all clear whether the omission of CsCl will give similar results in other soil types for example, highly mineral soils or high clay soils. The soils surveyed here are all similar in terms of their formation in tundra, with lots of organic content. Indeed, the authors state that samples were collected within 120 m of each other. So, as interesting as the results are, it is very difficult to know whether these methods could be applied to any other soils with the same result.

Fig. 5. I am not sure why the square root of copy number shown and not actual copy number. What is the reasoning behind this transformation?

·

Basic reporting

The is a well written, clear manuscript which deals with the generation of viromes from soil samples. Several common DNA extraction and purification methods are systematically evaluated against DNA yield, purity, successful metagenomic library preparation and assembled virus contigs. The finding will be very useful to anyone working in the field of virus metagenomics, as an optimum extraction method is identified for soil samples, and one common virome purification methods (CsCl purification) is shown to significantly reduce DNA yield and the number of virus contigs assembled compared to other methods. This study also addresses ssDNA and dsDNA virus ratios by using a relatively new library preparation kit to infer that dsDNA viruses make up the vast majority of particles in the soils tested.

Experimental design

see General comments

Validity of the findings

see General comments

Additional comments

L26 superscript on 10^7 and 10^9 didn't come through on my copy. Virus-like-particles or confirmed viruses?

L102 What were the units of DNase used?

L105 superscript on cm3

L197 - Can you label this header "Experiment 1: Different DNA extraction......" for clarity.

Figure 2A/B: Nanodrop absorbance ratios: I'm not clear how you managed to get reliable 260:280 ratios given such low DNA extraction concentrations shown on Figure 2A? They are often unreliable at such low concentrations. The total DNA yield would also be of interest in the text somewhere, I didn't see the volume written.

Figure 2C - You make a valid point (L213-214) that A260:280 is a better predictor of library success than A260:230 (which is the main reason for doing this study). Given you don't use the 260:230 ratios any further in the study I would consider removing Figure 2C. It is a busy figure and distracting with the wide error margins. Also, ANOVA would be preferable to doing multiple t tests on all the different combinations. Possibly the low DNA concentrations are throwing off these ratios and producing the high error values or some unimportant impurity is left in the sample. Either way, I don't think you need 2C as the success criterion is library generation which you deal with later on.

L217-218 "For Bog.....PowerSoil extracts consistently exhibited the highest A260:230 ratios (i.e. inferred to be cleanest)." - When I read Figure 2B, in Bog there is no significant difference between powersoil and the others? CTAB + Ampure fits the ratio limits of 1.6-2.1 best.

L255 Can you label the header "Experiment 2: Heat-based lysis...." so its easier to follow.

L273-274 "CsCl purification had the most impact: yield was higher without it than with it for all but one sample (Palsa, –CsCl[BB])" - Palsa - CsCl BB shows ND (non-detectable DNA concentrations) in Figure 4. "Without CsCl" treatments look consistently the best in all samples from Figure 4.

L291 on the viral signal

L302 - These are the 12 viromes from figure 4? 9 of these had no detectable DNA in them via Qubit. It might be worth mentioning then in L244 onwards that the library prep kit amplifies DNA from the picogram range.

Figure 7 - Can the C suffix be labeled "soil core" or similar in the legend to aid in understanding the groupings.

Figure 3 is not referenced or discussed in the text

---

## Round 0.2 · Minor Revisions

Thank you so much for your hard work with the manuscript. This is a really interesting paper and I think that the data are very useful. I did send it back to one of the reviewers and they were also very satisfied with the changes made and points of clarification. Nevertheless, there is still one point that requires some additional work. I agree with the reviewer that it would be really simple to add some fluorescence microscopy data/images to fully demonstrate the lack of non-viral contamination. I understand that this generates additional work, but it would also provide a strong evidence of the success of the method. In case the addition of the fluorescence microscopy data is impossible, could you, at least, incorporate the points made by the reviewer about it within the text? Then, we can always leave the reader to decide.

Best wishes

Alex

Reviewer 1 ·

Basic reporting

I am satisfied with the changes the authors have made in terms of improving clarity/meaning, and inclusion of additional references.

Experimental design

Likewise, I am satisfied with the edits made to address the previously stated concerns.

Validity of the findings

Thank you for the perspective on the nature of the peat soils - particularly the damage incurred by sampling. This is something that is often overlooked - or at least unmentioned - in field studies, and might be worth mentioning explicitly in this manuscript.

I am satisfied with the edits made, save one specific item.
With regard to confirming increases in DNA yield with (-) CsCl approaches (l. 270-290): You seem to insist that PCR screening is sufficient to check for non-viral contamination. I am not sure why you could not screen your viral extracts using fluorescence microscopy to check for the presence of cells (or cell-sized bodies). This would provide a level of confidence in the origin of the subsequently extracted nucleic acids that cannot be achieved with PCR alone. I really think this should be done (and example images provided in the supplemental materials) to help characterize what is in the (-) CsCl extracts. It is simple, cheap, and would quickly reveal the extent (if any) of bacterial contamination in these samples, and extinguish any lingering concerns from readers such as myself.

Additional comments

I appreciate your thoughtful consideration of and responses to my questions and concerns. It can be difficult work to endure criticisms, or to be asked to do even more, when so much effort has already been invested. Thank you for bearing with the process. I really think that in the interest of transparency and good reporting, you ought to perform fluorescence microscopy on the (-) CsCl samples and make the resulting data available in the manuscript. I look forward to seeing this manuscript in print after attending to this last concern.

---

## Round 0.3 · accepted · Accept

Dear Gareth

I really appreciate all the effort you have done with the manuscript. I feel convinced by your explanations regarding the methods used and I can also agree that microscopy is not the answer for everything. The important is that the reasoning for using different approaches is clear in the manuscript. Many thanks for bearing with us during the process. All the very best

Alex